# Fungicide Resistance in *Botrytis* spp. and Regional Strategies for Its Management in Northern European Strawberry Production

**DOI:** 10.3390/biotech12040064

**Published:** 2023-11-19

**Authors:** Roland W. S. Weber, Antonios Petridis

**Affiliations:** 1Lower Saxony Chamber of Agriculture, Esteburg Centre, Moorende 53, 21635 Jork, Germany; 2Department of Food Science, Aarhus University, Agro Food Park 48, 8200 Aarhus, Denmark; apetridis@food.au.dk

**Keywords:** *Botrytis*, fungicide resistance, grey mould, horticulture, IPM, northern Europe, nursery plants, strawberry

## Abstract

Grey mould, caused by *Botrytis cinerea* and other *Botrytis* spp., is a major cause of fruit rot in strawberries and other fruit crops worldwide. Repeated fungicide applications are essential in order to secure harvests. However, resistance to all currently registered single-site fungicides is widespread. The rising importance of strains with multiple resistance to most or all fungicides is of particular concern. These strains may be introduced into fields via contaminated nursery plants and/or by immigration from adjacent plots. On the basis of research conducted in northern German and Danish strawberry production, a concept to manage fungicide resistance under northern European conditions has been developed and put into regional strawberry production practice. This principally includes the testing of nursery plants for fungicide-resistant *Botrytis* strains prior to planting; the restricted and specific use of fungicides at flowering in the production fields, taking account of the resistance spectrum within the local *Botrytis* population; and crop sanitation measures such as the removal of rotting fruits at the beginning of harvest. Further options such as protected cultivation, reduced fertilisation and biological control are also discussed. The practical implementation of such a strategy in northern Germany and Denmark has been shown to reduce the occurrence of multi-resistant strains to a tolerable steady-state level.

## 1. Introduction

*Botrytis cinerea* Pers.: Fr. *sensu lato* is one of the most important fungal pathogens of a wide range of horticultural crops, including fruit crops such as strawberries and other soft fruits, cherries, plums, apples, and pears [1]. Because the rotting fruit rapidly becomes covered by a dense lawn of grey conidiophores with asexually produced spores (conidia), the disease is commonly referred to as ‘grey mould’. Among major fruit crops, strawberries are particularly prone to grey mould which causes high pre-harvest and even higher post-harvest losses [2]. There is a wide repertoire of mechanisms by which *Botrytis* infects its host’s organs [3]. On strawberries, the fungus overwinters on living and dead plant material (Figure 1a). Infections of living leaves may remain quiescent for weeks or months until the onset of leaf senescence, which triggers sporulation. Such spores produced on colonised dead plant material are a major source of inoculum for fruit infections [4,5]. Primary fruit infections (Figure 1b) are initiated at flowering, but the outbreak of the aggressive rot is delayed until the fruit begins to ripen [6,7,8]. The ripening process is the result of biochemical and physiological changes in the fruit, which favour *Botrytis* [9].

Once the first fruits are covered by grey mould, airborne conidia are released in great quantity by wind and rainsplash. These initiate secondary infections of further flowers and ripening fruit (Figure 1c). In addition, healthy fruits in physical contact to an infected one are attacked by mycelial overgrowth [10]. Such a polycyclic infection biologically dictates that the focus of fungicide sprays must lie in controlling the primary infections at flowering [2]. Later sprays are mostly ineffective, encourage the development of fungicide resistance on a local scale (see below), and give rise to problems with detectable fungicide residues in the harvested fruit.

Genetically distinct entities have repeatedly been partitioned off *B. cinerea sensu lato* and are now being regarded as separate species. A distinct species on strawberries is *B. pseudocinerea* Walker et al., which is morphologically identical to *B. cinerea* but differs in key biological aspects. It typically infects its hosts early in the season, being displaced towards harvest by other *Botrytis* spp. [11,12,13,14]. In commercial Danish and northern German strawberry fields, the share of *B. pseudocinerea* at harvest was around 5–10% in 2010–2019 [15,16]. In northwestern Europe and North America, *B. fragariae* Plesken et al. is a third species commonly found on strawberries [17], and yet further species have been reported elsewhere [18]. Even after these taxonomic rearrangements, there are further groups of strains with distinct fungicide resistance patterns, such as *Botrytis* group S, whose taxonomic status has not yet been finalised [19,20]. While *B. cinerea*, *B. fragariae*, and *Botrytis* group S are able to develop resistance to all of the currently used single-site fungicides (see below), *B. pseudocinerea* seems to be unable to do so [12]. This might partly explain its occurrence early in the season, i.e., before the first fungicide sprays [11,12].

In northwestern Europe as well as in other parts of the world, grey mould may cause the total loss of a strawberry harvest in wet seasons in unsprayed fields (Figure 1c). Fungicides are therefore paramount to ensuring a marketable yield, although substantial losses of 10–25% may still occur even if fungicides are sprayed repeatedly [21,22]. Further, an enhanced incidence of grey mould before harvest will compromise the storability and shelf life of harvested fruit. Under such conditions, organic production is not commercially viable at present because most consumers are not prepared to pay the higher price for perishable and highly seasonal crops such as organic strawberries. This is in contrast to other crops which can be subjected to long-term storage and are marketed all year round, such as apples. Strawberry production, according to the guidelines of the Integrated Pest Management (IPM), is a major branch of fruit production with acreages of approx. 4000 ha in northern Germany and 1000 ha in Denmark. Fruit farmers are heavily reliant upon effective fungicides to control grey mould. Fungicide-resistant strains of *Botrytis* spp. are posing a fundamental threat to securing stable yields and maintaining high quality standards.

Excellent reviews are available on specific aspects of fungicide resistance in *Botrytis* in relation to the underlying molecular mechanisms [23,24,25] and on selected non-chemical aspects of managing grey mould in strawberries and other host plants (references, see below). These have brought together worldwide research data on specific topics. In this article, we follow a different line by considering all relevant results from a geographically delimited area comprising northern Germany and Denmark. These neighbouring strawberry-growing regions share a similar climate and similar production methods. On the basis of earlier findings (reviewed in [26,27]) and recently gained knowledge, we are now able to propose an integrated concept of fungicide resistance management, which has been put into regional strawberry production practice. An outlook towards future options is also provided.

## 2. Fungicides and Fungicide Resistance

### 2.1. Multi-Site and Single-Site Fungicides

Multi-site (protectant) fungicides, such as captan or dithianon, interfere with basic aspects of the fungal metabolism and are therefore not generally prone to resistance development by *Botrytis* [23], although strains with reduced sensitivities to captan have been observed [28]. With the exception of tolylfluanid, which was banned from use in 2007, the efficacy of protectant compounds even against susceptible *Botrytis* strains is somewhat limited. Strains with a reduced sensitivity to captan may not be controlled by this fungicide under experimental conditions [28]. Captan is currently available to strawberry farmers in some countries (e.g., Germany) but not in others (e.g., Denmark).

In practice, therefore, control of grey mould is heavily reliant on single-site fungicides. Because these interact with defined molecular targets, they have a highly specific and effective mode of action, but they are also at high risk of resistance development by specific mutations [23,24,25]. In addition to these target-site mutations affecting individual fungicides or members of the same chemical class, there is a multi-drug resistance (MDR) mechanism based on the expulsion of selected molecules from the fungal cell. This is achieved by energy-driven efflux pumps located in the plasma membrane [17,19,29]. Such MDR mechanisms may act against several chemically unrelated molecules.

### 2.2. Resistance to Historic Single-Site Fungicides

The first single-site fungicides in fruit production were the methyl benzimidazole carbamates (MBCs) such as benomyl, carbendazim, or thiophanate-methyl. Within a few years of their release in 1969, resistant *Botrytis* strains were observed [30,31], and the use of these compounds had to be discontinued. MBC molecules bind to the β-tubulin subunit of fungal microtubules, thereby disrupting the cytoskeleton and hyphal growth polarity. Resistance is caused by mutations at codons 198 or 200 of the *Mbc1* gene, which encodes β-tubulin [32]. These mutations result in a complete loss of activity of MBC fungicides against resistant *Botrytis* strains. Even 50 years after the end of MBC use in strawberries, MBC resistance is still common in northern German and Danish *Botrytis* populations [15,16], indicating no competitive disadvantage of the mutants relative to the wildtype. Similar observations have been made elsewhere [33]. The most persistent mutations appear to be those at codon 198 [23].

The dicarboximide compound iprodione, registered in Germany in 1979–2009, suffered a similar fate as the MBCs [34,35] but was deployed for a longer period before being phased out. The precise mode of action of iprodione is unknown, although the stimulation of osmotic stress responses suggests a target in signal transduction [24]. Resistance to iprodione is based on mutations in the *bos1* gene (formerly named *Daf1*) which encodes a histidine kinase involved in cell signalling [36,37]. Resistance factors, i.e., the differences in fungicide inhibitory concentrations between resistant and sensitive isolates, are generally lower in iprodione than in MBC fungicides [38]. As with the MBC fungicides, resistance to iprodione has persisted in the environment after the end of fungicide use in northern European strawberries [15,16]. Most of these resistant strains showed moderate rather than high levels of resistance (R.W.S. Weber, unpublished).

### 2.3. Resistance to Currently Used Single-Site Fungicides

The turn of the millennium witnessed a spree of new fungicide releases against grey mould and many other diseases in fruit production. These included quinone-outside inhibitors (QoI) such as trifloxystrobin, pyraclostrobin, and azoxystrobin; succinate dehydrogenase inhibitors (SDHIs) such as boscalid and fluopyram; hydroxyanilides such as fenhexamid and (much later) fenpyrazamine; anilinopyrimidines (APs) such as pyrimethanil, cyprodinil, and mepanipyrim; and phenylpyrroles such as fludioxonil. We shall briefly consider these five groups in turn.

QoI fungicides bind to the cytochrome b molecule of the electron transport chain in the inner mitochondrial membrane, thereby blocking mitochondrial respiration. Point mutations are almost universally found at codon 143 (G143A) and cause a complete loss of activity [39] with very high (>1000-fold) resistance factors. In fruit production, the most widely used QoI fungicides are trifloxystrobin, pyraclostrobin, and azoxystrobin. There is complete cross-resistance between them, meaning that the G143A mutation will affect all QoI inhibitors [40]. The very widespread occurrence of QoI resistance in northern Europe [15,16] and critical laboratory experiments [41] argue against any impairment of fitness in G143A mutants in *Botrytis*.

Succinate dehydrogenase is an enzyme comprised of four subunits located in the inner mitochondrial membrane, catalysing the oxidation of succinate to fumarate within the citric acid cycle. Several point mutations in the *sdhB*, *sdhC*, and *sdhD* genes encoding different subunits of succinate dehydrogenase have been characterised [24]. These interact differentially with boscalid, fluopyram, or other SDHIs, meaning that there is only partial cross-resistance between them [42,43]. This leads to a wide range of resistance factors to SDHIs [42]. There is evidence from some studies [41,44] but not from others [45,46] that mutations causing resistance to SDHIs are associated with a reduced fitness of the strains harbouring them.

Fenhexamid is the most important member of the hydroxyalinide group of fungicides. In contrast to the other four main groups of current botryticides, hydroxyanilides have a very narrow spectrum of activity confined to *Botrytis*, *Monilinia*, and related fungi [47]. The molecular target is the enzyme 3-ketoreductase in ergosterol biosynthesis, encoded by *erg27* [48]. High levels of resistance are caused by a mutation at position 412 (F412) and moderate resistance levels by numerous other mutations in the same gene [49]. Fitness costs associated with F412 include a sensitivity to freezing, slower growth, and reduced spore formation [48], although these may not be decisive in the field [50]. *Botrytis pseudocinerea* is able to grow in a restricted and characteristically branched manner on rich agar media augmented with fenhexamid [13]. This species is viewed as possessing a modest degree of natural resistance to fenhexamid [11,23].

APs are an important group of fungicides with activity against *Botrytis* spp. and numerous other fruit pathogens. APs have long been considered inhibitors of amino acid biosynthesis, although the molecular target remains unidentified [23,24]. Very recent research points to a mutation in the *Bcmdl1* gene, the product of which is involved in mitochondrial ATP synthesis [51]. There is a high degree of cross-resistance between the three APs in common use [52,53]. There is no obvious fitness cost associated with AP resistance in *Botrytis* [54].

The phenylpyrrole fungicide fludioxonil has a similar mode of action to iprodione, but in this case, mutations in *bos1* with high resistance factors are extremely rare [55,56] and seem to play no relevant role in fruit production practice. Prohibitively high fitness costs may be the reason. Somewhat lower resistance factors to fludioxonil may be caused by MDR-type mechanisms of which several variants have been reported. These are also active against other fungicides, notably APs [57,58]. The efficacy of these fungicides against MDR strains may be severely curtailed under practical conditions.

## 3. Resistance Development over Time

### 3.1. Overall Frequencies of Fungicide Resistance

In northern German strawberry fields, resistance to the above fungicide groups, as well as the formerly used MBCs and iprodione, has been monitored annually for a decade, starting in 2010 [15,59]. Throughout this period, resistance to individual fungicides has been present at more or less constant and uncomfortably high levels, except for resistance to boscalid (registered in 2009) and fluopyram (registered in 2015) which showed a steady increase during that decade. The annual shares of resistance were below 10% (fluopyram), 10–40% (thiophanate-methyl), 25–60% (boscalid, cyprodinil, fludioxonil, fenhexamid, iprodione), or above 60% (trifloxystrobin). Similar data on shorter timescales were obtained for Denmark [16] and from several other countries and cultures [25,60,61,62].

A comparison of the levels of resistance to representatives of all five currently available fungicide groups in northern Germany and Denmark is given in Figure 2. However, a focus on individual fungicides masks an underlying trend towards the development of multi-resistant (MR) strains combining within them resistance to most or even all currently available fungicides (Figure 3). Similar trends towards MR strains of *Botrytis* in strawberry production are being observed worldwide [25,63,64,65].

### 3.2. Combinations of Fungicide Resistance

An analysis of resistances to different fungicides reveals that these are not randomly distributed among *Botrytis* isolates. Firstly, there is commonly a double resistance to compounds which are combined within the same commercial product. For example, while QoIs may be applied singly in northern European fruit production, the SDHI boscalid has been registered only in combination with the QoI pyraclostrobin under the trade name of Signum^®^. Thus, resistance to QoIs may occur singly, whereas boscalid resistance is nearly always combined with QoI resistance [14,59]. Similarly, in German strawberry production, APs are mostly applied as the combination product Switch^®^ (cyprodinil plus fludioxonil) which is registered for a maximum of three applications per season. In Denmark, Switch^®^ is limited to a single annual use, meaning that the APs pyrimethanil or mepanipyrim have been used singly as an alternative to Switch^®^. This situation is reflected by similar resistance levels to fludioxonil and APs in Germany but elevated AP resistance relative to fludioxonil in Denmark (Figure 2) [16].

Whilst local, regional, or national *Botrytis* populations as a whole seem to adapt to prevailing fungicide regimes [26,27], not all strains do so equally readily. Thus, *B. pseudocinerea* as well as *B. cinerea* strains with resistance to zero or single fungicides are more common in untreated or organically managed fields and orchards [13,14]. An analysis of the number of fungicide resistances per isolate revealed that strains harbouring resistance to many or all of the five currently registered fungicide classes were far more likely to possess resistance also to the historic compounds, MBCs and iprodione, than strains with no or few resistances to current fungicides (Figure 4) [15,16]. This pattern is best explained via a sequential acquisition of fungicide resistances by competent strains, proceeding in the direction MBC → iprodione → QoI → SDHI [27,66]. A similar scheme, albeit with different fungicides and a different sequence of resistance development, has been proposed for the apple scab fungus, *Venturia inaequalis* [67].

## 4. Selection of Fungicide Resistance in the Field

In forced-selection experiments, shares of resistance to individual fungicides in the same field are determined before and after a sequence of blossom-time fungicide sprays. The results from such trials leave no doubt that a selection of resistant strains can take place even within one season [9,63]. Forced-selection experiments are particularly convenient with raspberries because *Botrytis* is readily isolated from overwintered fruit mummies and infected canes before flowering in spring, and again from ripening fruit in summer [26]. In strawberry trials, we were able to detect similar effects even in very small plot sizes of a few square metres (R.W.S. Weber and A.-P. Entrop, unpublished).

In terms of fungicide resistance levels in different fields, the cumulative number of sprays is a more accurate parameter to determine fungicide selection pressure than field age because it accommodates the intensity of spraying. We were able to compute such data for Denmark because past spray schedules were made available by all farmers at the time of sampling [68]. These data confirm a fair relationship between the number of applications of all single-site fungicides—counting a single spray with a tank mixture of two compounds as two applications—and the average number of fungicide resistances per isolate (Figure 5). Similar data were obtained for sweet cherry orchards in northern Germany [14]. At least in short-lived cultures such as strawberries, there is always a residual error in such computations because the level of resistance in strains of *Botrytis* introduced with the planting material may differ between fields, and this cannot be assessed retrospectively. The key contribution of planting material to resistance levels in production fields is described below (Section 6).

## 5. The Origin, Spread, and Further Fate of MR Strains

MR strains are potentially dangerous to fruit production because they challenge recommendations based on good agricultural practice and IPM guidelines, which state that successive fungicide sprays should include different chemical classes in order to prevent resistance development [69]. Following such a strategy might actually create a selective pressure favouring MR strains [26,70]. In this sense, the increasing frequency of MR strains in strawberry fields requires a reconsideration of IPM principles. It is therefore pertinent to ask how MR strains colonise strawberry fields.

### 5.1. Routes of Entry of MR Strains into Strawberry Fields

The world’s first MR strains must have arisen de novo by a sequential accumulation of resistances to different fungicides. This might have occurred on several separate occasions [66]. It is possible that such founder strains developed in established northern German raspberry fields in about 2012 [60]. Alternatively, the long-distance import of a strain followed by local spread cannot be ruled out. Such a case has been reconstructed for the migration of a genetically distinct strain from French to southern German vineyards [71]. Either way, the most likely explanations of the mass appearance of MR strains in commercial fields are the introduction events of MR strains with contaminated nursery plant material or the immigration from adjacent fields, followed by their local selection due to fungicide use. There are examples of both these routes in northern Germany.

Both strawberry and raspberry nursery plants have been demonstrated to be heavily colonised by MR strains, whereby some nurseries were much more strongly affected than others. For some plant batches, shares of 50% MR strains among all recovered *Botrytis* isolates were recorded [65,72,73,74]. In raspberries cropped for several years in open-field culture, there was an increase in the share of MR strains within the total *Botrytis* population after the first spraying season relative to their frequency at planting [73]. Thus, MR strains can be introduced with planting material, and they can still be detected in subsequent years, gaining local dominance if the selection pressure imposed by intensive fungicide use is high.

Long-term observations of the same fields have provided evidence of the introduction of MR strains by short-distance immigration. When a new plantation of raspberry long canes heavily contaminated with MR strains was set up alongside a long-established field with a history of consistently low annual resistance levels, a massive rise of MR strains was observed in the latter field within one season (Figure 6). Similarly, once MR strains had invaded a northern German IPM cherry orchard originally free from them, they were recorded in each of the several successive years until finally disappearing again [14]. In another case, the migration of MR strains from a strawberry field to a nearby vineyard in southern Germany could be reconstructed [60].

### 5.2. Competitive Fitness of MR Strains

At least as conspicuous as the high frequency of MR strains in intensively managed strawberry and raspberry fields is their near-absence from unsprayed plants even within a few metres of the field boundary (Figure 7). This phenomenon, which we have observed repeatedly, is a possible part-answer to the question concerning the environmental fitness of MR strains. It suggests obvious fitness deficits in the absence of the selection pressure imposed by the repeated use of single-site fungicides. Mutations leading to fungicide resistance are a deviation from the wildtype, and the question about associated fitness costs is complex. Fitness costs due to individual mutations are low or absent in the case of MBCs, QoIs, APs, SDHIs, and fenhexamid, at least under laboratory conditions (see Section 2.3). Even strains combining resistance to numerous fungicides often fail to show appreciable deficits in growth, osmotolerance, sporulation, virulence, or other measurable parameters in the laboratory, in comparison to wildtype isolates [60,75].

The fate of MR strains in the field is of greater interest to fruit growers. There are contradictory results. In the USA, observations in an artificially inoculated blackberry field indicated that MR strains were able to compete with the native population both with and without selection pressure imposed by fungicide use [76]. In contrast, in our own experience of the dominance of MR strains in intensively treated fields (including nurseries), their gradual decline over several years in fields or entire regions implementing reduced spray programmes, and their near absence elsewhere, suggests a reduced ability to compete with wildtype strains in nature. Experimental results from mixed-inoculation experiments with or without fungicide use point to a similar direction [77,78]. Such obvious differences in the competitive ability of MR strains in various studies might reflect different experimental approaches or differences in the genetic composition of the strains used. In northern Europe, reduced competitiveness could be key to unlocking options for managing fungicide resistance in practice (see below).

## 6. The Practical Relevance of Clean Nursery Material

As we have seen above, nursery plants may be contaminated by *Botrytis* strains—including MR ones—at the point of delivery to the fruit farmer. This raises key questions about the practical relevance of such contaminations. In order to examine this aspect, experiments were conducted in which batches of strawberry plants obtained from different nurseries were examined at the point of delivery to the farms, and batches differing in overall levels of *Botrytis* contamination and/or in the shares of MR strains were chosen for explanting on an experimental field at Aarhus University. All plants were treated with the same fungicide regime comprising a standard spray sequence of Signum^®^ (pyraclostrobin + boscalid), Switch^®^ (cyprodinil + fludioxonil) and Teldor^®^ (fenhexamid) at flowering. On all plants, all *Botrytis*-infected ripe fruits were collected and analysed for fungicide resistance in the first cropping season. Secondary infections were prevented by frequent fruit pickings. The results indicated that plants with low levels of *Botrytis* infections developed low levels of grey mould at harvest. Similarly, plants more heavily infected by strains with resistance to only one or a few of the fungicides developed little grey mould at harvest. In contrast, plants harbouring MR strains at the nursery stage retained these in the field, whereby the level of grey mould at harvest was correlated to the abundance of MR strains in the nursery plants [61]. Two illustrative examples of these results are given in Figure 8.

## 7. Synthesis: A Practicable Concept for Managing Fungicide Resistance

*Botrytis* isolates identified in our laboratory tests as being resistant to any or all of the existing fungicides have been repeatedly shown in inoculation experiments to be able to infect strawberry and other fruits pre-treated with the respective compounds in a manner indistinguishable from control plants sprayed with water only [59,60]. In contrast, sensitive isolates were fully controlled by the fungicides in the same experimental setups. Similar findings have been made elsewhere [64]. We must therefore accept that fungicide resistance in *Botrytis* is a real threat to horticulture rather than a laboratory artifact. This renders the effective management of fungicide resistance an essential task.

### 7.1. Nursery Material

As we have seen, the extent of fungicide resistance in a given field—and especially the share of MR strains—is mainly a function of nursery plant contamination and the chosen fungicide regime. Experiments with nursery plants bearing different levels of fungicide-resistant strains have revealed that the main attention should focus on MR strains. A maximum contamination of 5–10% of plants with MR strains has been proposed as an upper threshold for successful strawberry production under northern European conditions which usually feature two cropping seasons [79]. In order to assess nursery plant contamination, a practical test has been developed and made available to farmers and their advisors. This test requires only simple laboratory equipment. It is based on the incubation of 50 plants per batch in a shallow damp tray at 4 °C or at room temperature for 1–2 weeks, followed by the isolation of all visible grey mould colonies onto a suitable agar medium. Resistance tests with conidia from these agar plates can be conducted in the same manner as with sporulating fruit at harvest [38,43].

Future work must carry these findings into the plant nurseries, which should be encouraged to implement a quality control scheme, leading to certifications of batches free from MR strains. In order to achieve these goals, nurseries must avoid using the same fungicides that are being deployed for grey mould control in the production fields because, as we have seen above, it is the cumulative use of the same set of fungicides during the entire lifetime of a plant which facilitates the buildup of MR populations. New approaches may provide a solution to this problem, such as hot water treatments of strawberry nursery plants to eliminate *Botrytis* infections [80,81]. These could be conducted prior to the delivery of plants to the farmer or on-farm.

### 7.2. Intensity of Fungicide Use

Irrespective of the degree of contamination of nursery plants, farmers should be aware that a more intensive fungicide regime will lead to an accelerated buildup of fungicide resistance in their fields. Deceptively, yield reductions are often overlooked in one-year efficacy trials such as those normally conducted by research stations. This is because MR strains selected in the first season of fungicide sprays and present at the first harvest will only be available for infection in the second season. Thus, whilst a more intensive spray sequence may give rise to a higher overall control level in the first year, it also causes an enhanced enrichment of resistant strains which will jeopardise harvests in subsequent years. In the longer term, therefore, fewer annual sprays will give higher overall yields, as illustrated in Figure 9 [82].

This theoretical notion has been tested in practice somewhat serendipitously. In the northern German state of Lower Saxony, there were traditionally two separate advisory regions with historic differences in their approach to fungicide recommendations on strawberries. In region B, declining efficacies of fungicides were noticed as early as 2008. This led to the development and implementation of fungicide resistance tests on a regional basis [38,83], which uncovered widespread fungicide resistance. The results were communicated to the farmers and their consultants from late 2010 onwards. In response, the regional advisory service began to recommend a restricted number of three (exceptionally four) fungicide applications, confined to flowering, as from the 2012 season. In contrast, in region A, the traditionally high spraying intensity of five to six applications was continued until 2015. Comprehensive surveys in both regions in 2014 and 2015 showed strong differences in the abundance of MR strains which was high in region A but very low in region B (Figure 10). These data are highly relevant because farmers in both regions were being supplied by the same nurseries, thereby excluding the possibility of a differential introduction of MR strains with young plants. Such findings support an average of three sprays as being a practicable and sustainable resistance management strategy [27]. Similarly, in Denmark, the detection of extremely high levels of fungicide resistance in 2015 led farmers to reduce the number of sprays from five to six to about three to four per season, coinciding with a measurable decline in resistance levels by 2021 [16,68]. The practicability of this advice was confirmed by on-farm trials in northern German fields with elevated levels of MR strains in region A, where the unsprayed control showed a massive incidence of grey mould, whereas three fungicide sprays gave good control and six sprays gave no improved control (R.W.S. Weber and A.-P. Entrop, unpublished).

### 7.3. Disease Forecasting and Precision Agriculture

The above reduction in fungicide use may have been progressive in terms of saving on the number of sprays and thereby achieving a reduction in the share of MR strains, but it still relied mainly on calendar sprays at fixed dates (e.g., 10%, 25%, and 50% of flowers opened). Further progress lies in the more precise timing of sprays according to infection conditions. Several forecasting models have been developed for *Botrytis* on strawberries. The most important meteorological parameters are leaf wetness duration and temperature [84,85].

Any given model must be established in each growing region by adapting it to registered fungicides and to climatic conditions. Further, consultants and farmers alike must be convinced of the merits of such a system and be trained to use it. An advanced IT-driven system to pinpoint fungicide applications, the Strawberry Advisory System (SAS), has been developed and validated over several years in Florida [22,86] and was subsequently tested in other regions [87]. In comparison to calendar-based schedules, the SAS system reduced fungicide sprays by about 50% while restricting grey mould to similar or even lower levels, thereby increasing yields and profits. We have no data concerning the effect of this model on fungicide-resistant *Botrytis* strains. However, it can be expected that a massive reduction in the number of sprays would at least delay the rise of MR strains. Such an approach could therefore add a new dimension to fungicide resistance management.

## 8. The Contribution of Non-Chemical Approaches

There are good reasons to broaden our horizon concerning grey mould control beyond the use of fungicides. For one, the ailing fungicides need to be protected from further loss of efficacy, which is the consequence of resistance development. In the early days, they also occasionally needed protection from their own producing companies which promoted multiple successive sprays with the same (=their) compounds to maximise short-term profits. Secondly, ever higher hurdles have to be overcome in order to gain registration of new or re-registration of established fungicides. Therefore, we must prepare ourselves for a time when there will be fewer effective fungicides around. There are several non-chemical approaches which could augment or replace chemical fungicides. These are discussed below.

### 8.1. Cultivar Susceptibility

So far, there has been no reported case of monogenetic resistance of strawberries to infections by *Botrytis* [88]. Therefore, the breeding of new strawberry cultivars for enhanced grey mould resistance might ultimately give us more robust rather than fully resistant cultivars [9]. These could provide a more solid foundation for effective chemical and non-chemical approaches than the current cultivars, many of which are highly susceptible in northern Europe. Fruit firmness has been shown to correlate inversely with the incidence of grey mould in ripe fruits [89]. A test method for grey mould susceptibility based on the inoculation of ripe fruits has been developed. This test revealed large differences between cultivars, which were reproducible under field conditions [90]. The levels of compounds with antimicrobial activity such as anthocyanins also correlate with resistance to grey mould [91]. These, too, can be manipulated by breeding [92]. However, in fruit production, the marketing strategies and consumer preferences are based on aspects other than disease resistance or indeed almost any other factor concerning the practicalities of crop protection. In the short to medium term, the highest chances of acceptance of a new robust cultivar may lie in organic production, where such cultivars may make the difference between the non-feasibility and feasibility of strawberry production.

Everbearing (day-neutral) strawberry cultivars prolong the marketing period into late autumn. They are low in crop volume but attractive especially to farms with market stalls or on-farm shops. In such cultivars, the flowering period (and thus the time of fungicide applications) overlaps substantially with the harvest period. In northern Germany, such cultivars are subjected to more fungicide sprays than the standard spring-bearing cultivars with a shorter flowering period. Further, sprays may be applied when mouldy fruits are already present at high numbers. In consequence, everbearing cultivars are a hot spot for MR strains. In a survey in 2010 and 2011, the average share of MR strains among all examined *Botrytis* isolates was 48.0% in everbearing cultivars but only 6.7% in standard cultivars produced on the same five farms (R.W.S. Weber, unpublished data). Therefore, one strategy to reduce the spread of MR strains could be to breed autumn-bearing cultivars with shorter or more clearly delimited flowering and cropping periods. Additionally, particular attention should be given to autumn-bearing fields in terms of sanitation and biological control (see Section 8.3 and Section 8.4). Since everbearing strawberries contribute only a small share of the total acreage, a pragmatic solution could be their protected cultivation [93].

### 8.2. Protected Cultivation

There is a strong trend towards the protected cultivation of strawberries in glasshouses or tunnels in northern Europe and elsewhere. In northern Germany, the share is still low, but in the UK, protected cultivation has become the predominant form of strawberry growing [94]. The reasons include a more reliable yield, improved fruit size and shape, and earlier ripening at more favourable prices [93,95]. With respect to grey mould, epidemics in tunnels are often much less severe than in the open field, even in critical experimental comparisons [96,97]. In fact, the difference may be so strong that fungicides against grey mould may not be necessary at all in a well-managed tunnel [94,96]. Shorter periods of surface wetness, different temperature regimes, and a generally lower availability of infectious conidia may contribute to the reduced incidence of grey mould before harvest [94]. On the other hand, post-harvest grey mould can still occur in fruit batches from protected cultivation, meaning that the focus has to be on optimising storage conditions (see Section 8.5). Overall, protected cultivation can lead to the effective saving of fungicide sprays, but it has its drawbacks in raising production costs and giving rise to greater problems with powdery mildew and insect and mite infestations [97,98].

### 8.3. Aspects of Cultivation and Sanitation

There are several management options based on good cultural practice [99]. If secondary infections have their origin in the first ripening fruits developing grey mould, it makes sense to remove these during the harvest period in order to retard the onset of the exponential phase of the epidemic [100]. In the absence of concrete trial-based data for northern Europe, we recommend the collection and removal of mouldy fruit primarily during the first few harvest pickings, which are still low in volume, so that additional labour costs are a lesser issue than during the main pickings. This approach is now widely practised in northern Germany (Figure 11). These and other hygiene measures such as the removal of dead foliage or the disinfection of production facilities are more easily implemented in greenhouses or tunnels than in the open field [4,25].

*Botrytis* infections require high moisture levels or free water [6,7], and the incidence of grey mould can be reduced by promoting a rapid drying-off of plants. This means a wider spacing between plants [101], drip irrigation instead of overhead irrigation [102], and ventilation or heating to reduce moisture levels in greenhouses, tunnels, or field covers [103,104].

It seems an act of common sense to apply a reduced fertilisation regime with respect to nitrogen [2] in order to reduce vegetative growth to a necessary minimum. The reasoning is that excessive nitrogen fertilisation will yield fast-growing plant tissue which is often more susceptible to grey mould and other diseases, and that a dense canopy of foliage will prolong moisture periods on the flower or fruit surface [103,104]. In contrast to nitrogen, enhanced calcium levels have a grey mould-reducing effect in strawberries and other fruit [105].

### 8.4. Non-Chemical Crop Protection

Non-chemical alternatives to the chemical fungicides discussed in this review clearly have potential in controlling grey mould. These include biological control agents (BCAs), plant extracts, and other compounds registered for use in organic production. This vast subject has been reviewed in great depth [105,106,107], and we can only give a few selected examples here to indicate potentials and limitations.

Serenade^®^, a product containing *Bacillus amyloliquefaciens* (formerly *B. subtilis*), is one of the most widely characterised products, having been on the market since the turn of the millennium. More recently released products include Sentinel^®^ (*Trichoderma atroviride*), as well as several biofungicides containing *T. harzianum* or the yeast *Aureobasidium pullulans*. Any one of these may possess several modes of action against the same or different diseases, notably antibiosis, the induction of plant resistance, and competition for nutrients or space [105,107,108,109,110].

Trials with such ‘biologicals’ have shown particular promise under controlled conditions in the glasshouse or tunnel [105,106,111]. An interesting approach is the use of honeybees to deliver the inoculum into strawberry flowers [112,113].

In the open field, the efficacy of biocontrol agents is notoriously erratic. Efficacies have matched those of chemical fungicides in some studies [114,115] but failure has occurred elsewhere, highlighting the challenge to generate reproducible results. For example, when strawberry plants in open-field culture were treated with three different BCAs, *viz. B. amyloliquefaciens*, *A. pullulans*, and the filamentous fungus *Beauveria bassiana*, combined treatments were usually more effective than any single BCA, although the effective combinations varied between years [116]. Similar results have been obtained with mixtures of the bacterium *Bacillus mycoides* and the yeast *Pichia guilliermondii* [109,117]. However, in a different study on detached strawberry leaves, combinations of products containing *B. subtilis*, *T. atroviride*, and *T. harzianum* performed less well than any single product [118].

These examples demonstrate the highly variable nature of biological control in strawberries. Not surprisingly, no single product has as yet proved a full alternative to chemical fungicides, at least not under the conditions of open-field strawberry production in northern Europe. As a general rule derived from field trials in fruit production, the less favourable the conditions for a pathogen, the higher the efficacy of its control. Therefore, the current view is that biologicals should complement rather than replace chemical fungicides, e.g., by being applied in periods of suboptimal infection conditions or at additional spray dates before harvest and before or during flowering [117,119]. The application of biologicals before or even after harvest is particularly attractive because there are no issues of detectable pesticide residues to be considered.

There are several intriguing aspects with biologicals, highlighting the interconnections between seemingly unconnected parameters. For example, a reduction in N fertilisation in strawberries not only reduces grey mould levels but also seems to enhance the performance of *B. subtilis* [120]. In another context, there is evidence of differential effects of biologicals on specific *Botrytis* strains (reviewed in [105]). This point should be pursued further with respect to MR strains; if a reduced fitness of such strains rendered them more susceptible to biologicals, either alone or in combination with fungicides, the real usefulness of such products would be much higher than currently appreciated on the basis of simple trials. There is a high degree of speculative and wishful thinking in all of this, but that seems to be an integral part of biological control.

### 8.5. Storage

There is a considerable post-harvest component of grey mould. The outbreak of latent infections after harvest is strongly influenced by successful pre-harvest control measures and by storage conditions. Temperatures close to 0 °C greatly reduce storage grey mould. Harvested fruit should be cooled down as rapidly as possible [121,122]. A modified atmosphere—reduced O_2_ and increased CO_2_ levels—can further stabilise the harvested fruit during storage [123,124]. An enrichment of the storage atmosphere with ozone or a short ozone treatment of the fruit before storage can also reduce the incidence of grey mould in storage [125,126].

The treatment of harvested fruit before being placed in the storeroom has the potential to reduce the incidence of grey mould and other diseases, thereby prolonging storage periods and shelf life [105,106,114]. Bacteria and yeasts covering the fruit surface are the focus of intensive research, as are essential oils and other plant products, UV light, or treatment with heat or disinfecting agents (reviewed in [122,127]).

In many of these approaches, the observed suppression of grey mould and other rots may be due to a combination of a direct effect of the agent itself and an indirect effect via the stimulation of the immune system of the fruit [127]. The relevance of both factors has been demonstrated for the suppression of apple fruit rots by hot water treatment at harvest [128]. In strawberries, the effect of UV light as a trigger for the upregulation of genes encoding pathogenesis-related proteins has been characterised [129]. Similarly, pre-harvest sprays with chitosan, an inductor of plant resistance, may result in reduced grey mould incidence and an extended storage life [121].

## 9. Potential Environmental Impact

A specific assessment of the environmental impact of different strategies of grey mould management in strawberries is fraught with uncertainty and, to our knowledge, has not yet been attempted. On the one hand, individual fungicide sprays are conducted not only against *Botrytis* but also collaterally or chiefly against other fungal pathogens such as *Colletotrichum* spp. causing anthracnose or *Podosphaera aphanis* causing powdery mildew. On the other hand, other chemicals are not sprayed against *Botrytis* but against other foliar pathogens, soilborne organisms, or insect pests. Therefore, even a generalised agrochemicals factor in an impact assessment would not be a reliable indication of the environmental costs of *Botrytis* control. In very broad terms, however, it is fair to say that the application of biological control agents and/or the use of alternative post-harvest strategies will greatly reduce adverse environmental impacts, even if a quantification of this effect is not possible at present.

On a coarser scale, life cycle assessments have revealed a major difference between protected cultivation in tunnels or glasshouses and open-field cultivation, and between organic farming and IPM. Depending on the variables taken into account, soilless cultivation in tunnels has been calculated to generate the lowest environmental footprint in Spain. In combination with IPM, this approach produced the best overall ranking, with organic production ranking second best if the low tonnage per hectare was taken into account but better than IPM if this variable was excluded [130]. In a similar study of Swiss and Italian strawberry production, chemical soil sterilisation was the largest factor contributing adversely to the environmental impact, but, interestingly, longer cropping cycles also contributed adversely because of the generally low yield in the third year [131]. This latter point is of interest also in terms of grey mould control, given that MR strains tend to build up during the lifetime of a plantation. When the product environmental footprint was calculated for German and Latvian strawberries, greenhouse cultivation had the highest environmental impact followed by organic open-field production, whereas IPM open-field production and polytunnels produced the lowest environmental impact rating [132].

Differences in methodology may obscure the issue, but the outlines of the picture emerging at this stage indicate that polytunnels seem to hold potential both for a reduction in fungicide sprays against grey mould and, somewhat surprisingly, for a reduced environmental impact. A higher yield per unit of production area is the main reason why IPM fares better than organic production in such analyses under the current circumstances.

## 10. Conclusions and Further Work

In our understanding, the concept of IPM has always been intended to be a dynamic system of successive improvements aimed at restricting the use of synthetic crop protection compounds and other adverse environmental impacts while maintaining stable yields. The threat of fungicide resistance in *Botrytis* takes the guise of MR strains. This threat should be seen as an incentive for developing IPM to a higher level. Several approaches have emerged to reduce our dependence on chemical fungicides against grey mould. These are driven by concerns over fungicide resistance in the present review, but fungicide residues in the crop or lack of future fungicide registrations point to the same direction. The main conclusions and future areas of research are as follows:1.Nursery material should be free from MR strains. Ideally a quality control scheme for nurseries should be established, especially where few businesses supply large numbers of farms or even entire production regions.2.The use of chemical fungicides against *Botrytis* must be restricted to flowering, and the number of sprays should be limited. The use of disease forecasting models can further reduce the number of sprays and optimise fungicide efficacy by pinpointing suitable spray dates. The main aims are to retard the spread of MR strains, thereby maintaining fungicide efficacy at a high level.3.There is still some way to go before effective biological control measures are ready for large-scale strawberry production in open-field culture. Further research under given regional conditions is necessary. Disease forecasting systems may help to optimise application dates. It is expected that the efficacy of BCAs will improve if the available inoculum can be reduced or infection conditions can be altered.4.Hygiene and sanitation measures, such as the removal of dead leaves and rotting fruits at harvest, may reduce inoculum, but critical experiments on their efficacy are necessary under northern European conditions before labour-intensive measures can be recommended on a large scale.5.The effects of cultural measures such as reduced plant spacing or reduced nitrogen fertilisation also need to be critically evaluated under the cultivation conditions prevalent in any given area.6.Protected cultivation presents a major advance against grey mould. Little is known about the number and types of fungicide applications. Presumably, there is potential to reduce sprays. Market forces will dictate to which extent this form of cultivation is profitable in any given region. It is possible that the withdrawal of fungicides from registration will favour protected cultivation.7.The need to achieve rapid cooling of strawberries at harvest is paramount for post-harvest stability. Especially on hot days, the fruits should be collected several times per day and placed in the cold store. There is a need for research on long-term storage conditions for northern European cultivars and marketing situations.8.The breeding of more robust cultivars is a long-term goal. Such cultivars must be tested under regional conditions. Unfortunately, the decision about their introduction into the market is not taken by plant pathologists.

## Figures and Tables

**Figure 1 biotech-12-00064-f001:**
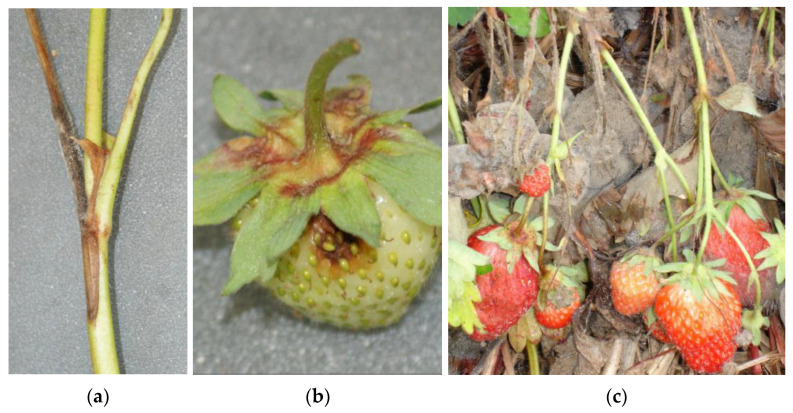
Grey mould on strawberries. (**a**) Infection of a dead inflorescence. (**b**) Primary fruit infection breaking out of latency at the onset of maturation. (**c**) Secondary fruit infections in an untreated field.

**Figure 2 biotech-12-00064-f002:**
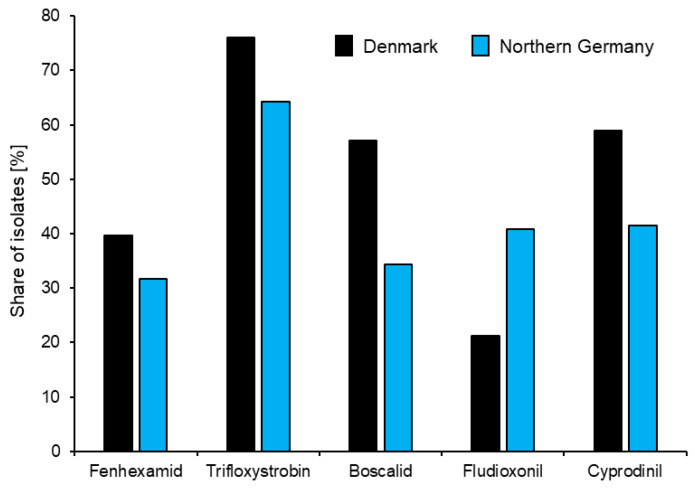
Comparative share of fungicide resistance among *Botrytis* isolates collected from strawberry fields in Denmark (46 fields; 594 isolates) and northern Germany (78 fields; 1019 isolates) in 2015–2019. Data from [16].

**Figure 3 biotech-12-00064-f003:**
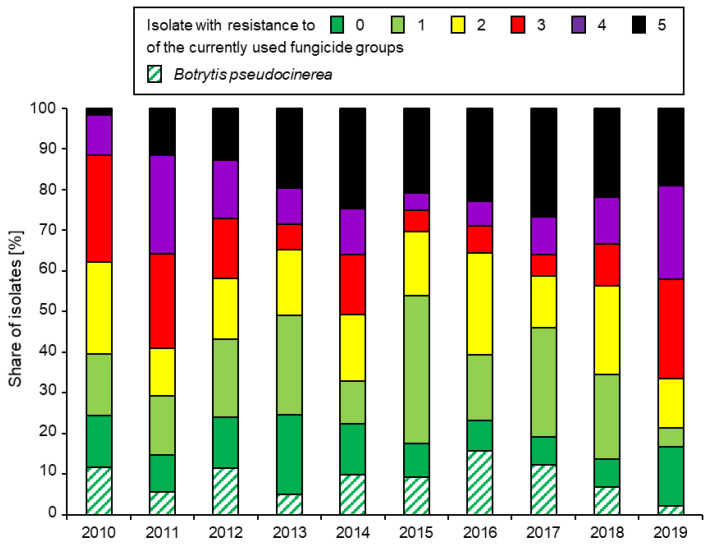
Annual share of northern German *Botrytis* isolates with resistance to 0, 1, 2, 3, 4, or all 5 of the currently used fungicide groups in 2010–2019. The share of *B. pseudocinerea*, identified on the basis of morphological details in the fenhexamid resistance test [13], is indicated as a subset of the fully sensitive isolates. Data from [15].

**Figure 4 biotech-12-00064-f004:**
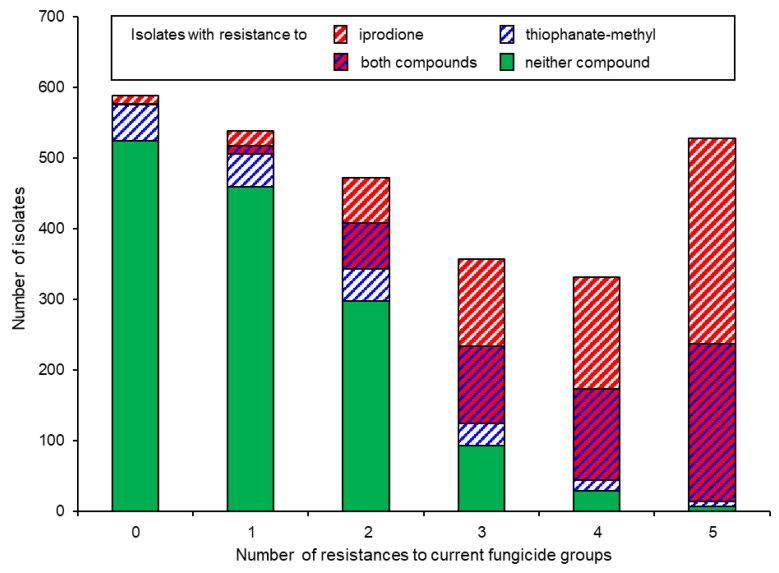
Occurrence of resistance to iprodione and thiophanate-methyl among isolates with resistance to zero or any up to all five currently used fungicide groups. Pooled data from northern Germany 2010–2019 [15].

**Figure 5 biotech-12-00064-f005:**
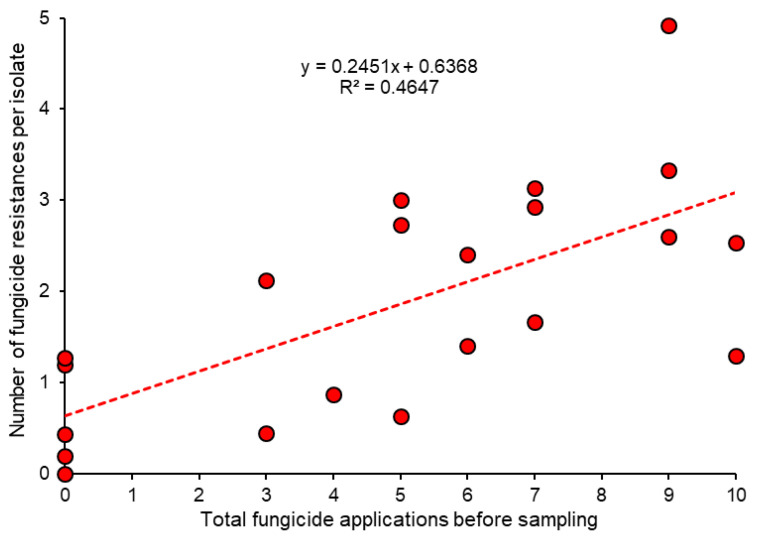
Relationship between the cumulative number of fungicide applications in Danish strawberry fields and the average fungicide resistance of isolates in each field. Data from [68].

**Figure 6 biotech-12-00064-f006:**
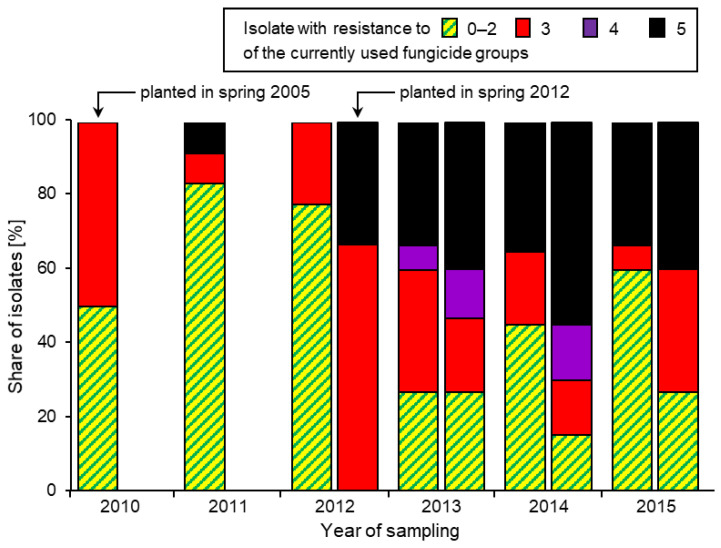
Share of *Botrytis* isolates with resistance to 0–2, 3, 4, or all five of the current fungicide groups, recorded annually on overwintered raspberry canes in an established field colonised by a *Botrytis* population with low levels of resistance (l.h. column) and a new spring 2012 plantation heavily contaminated with MR strains. Data from [73].

**Figure 7 biotech-12-00064-f007:**
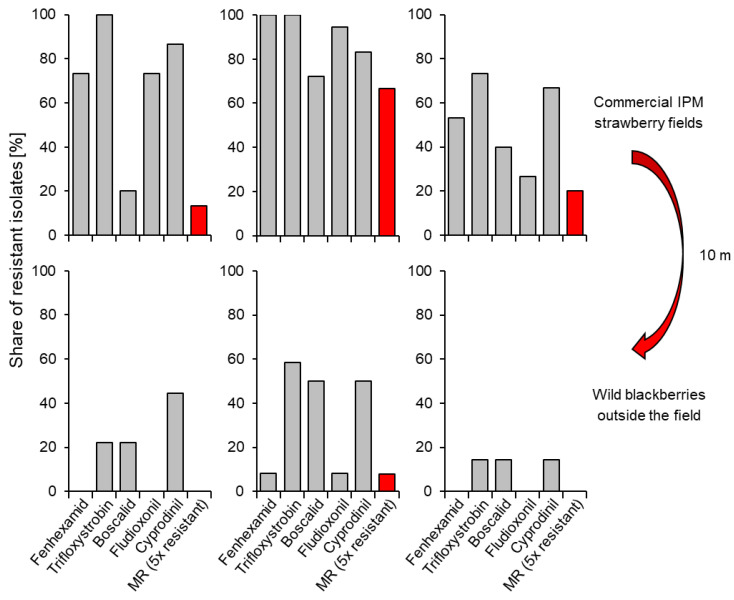
Shares of isolates of *Botrytis* with resistance to any one (grey columns) or all five (red columns) commonly used fungicides on fruits of three intensively sprayed strawberry fields and on wild blackberry bushes about 10 m beyond the perimeter of the field, i.e., just outside spray drift range. Approx. 12–20 grey mould isolates were analysed per sample (R.W.S. Weber, previously unpublished).

**Figure 8 biotech-12-00064-f008:**
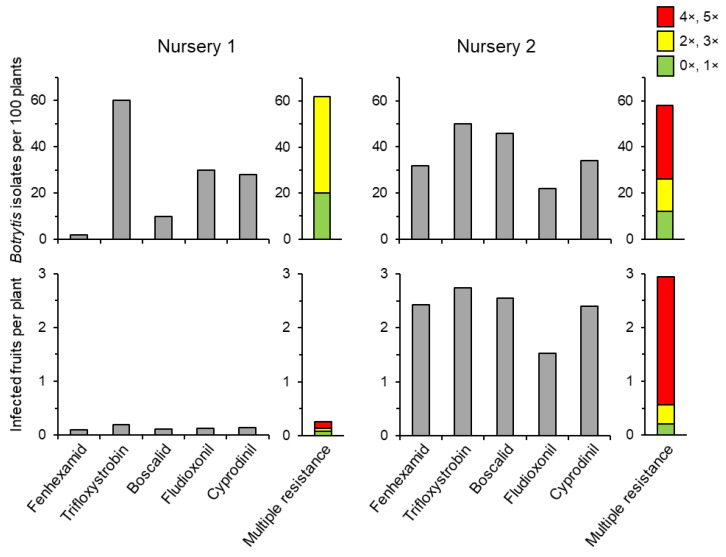
The effect of contamination of strawberry nursery plants with *Botrytis* (**top panels**) on the incidence of grey mould at the first harvest after a standard fungicide spray sequence (**bottom panels**). Results are given as numbers of isolates with resistance to each tested fungicide (grey columns) and to multiple fungicides (coloured column). From [79].

**Figure 9 biotech-12-00064-f009:**
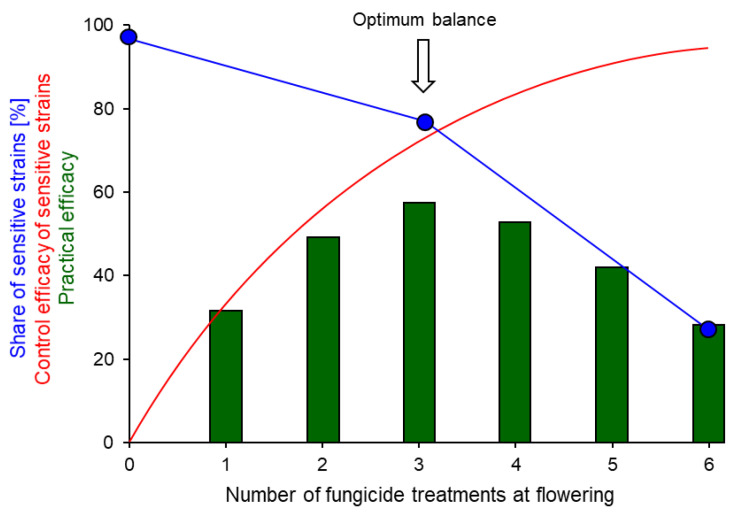
Relationship between the theoretical total efficacy of an increasing number of fungicide sprays at flowering against sensitive strains of *Botrytis* (red curve), the decreasing share of sensitive strains in the local population as a result of the selection of resistant strains by increasing fungicide treatments (blue line), and the resulting practical efficacy of spray regimes of different intensity (green columns). From [82].

**Figure 10 biotech-12-00064-f010:**
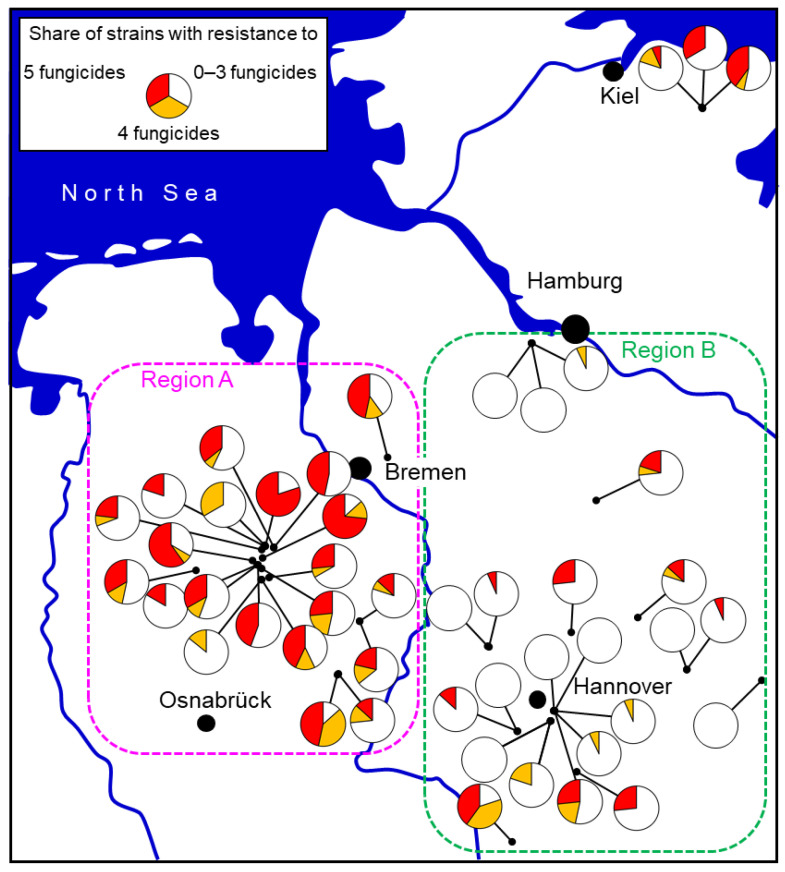
Share of *Botrytis* isolates with resistance to 0–3, 4, or all 5 fungicide groups in a survey of commercial strawberry fields in northern Germany 2014–2015. Two regions differing in the advice given to farmers concerning the use of fungicides also differed in resistance levels. In region A, 5–6 fungicides per season were commonly applied, whereas a more restrictive use of 3 sprays was recommended in region B. From [27].

**Figure 11 biotech-12-00064-f011:**
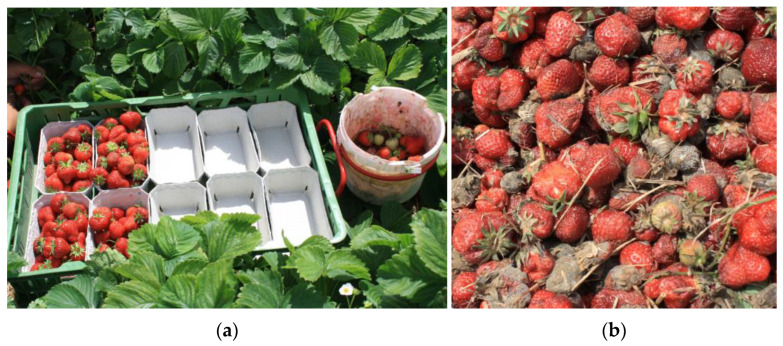
Crop hygiene during fruit picking. (**a**) Harvest cart with a tray for marketable strawberries and a bucket for infected and other damaged fruit. (**b**) Discarded fruit including numerous *Botrytis*-infected ones.

## Data Availability

Original data generated by the authors’ research are available in the articles cited or from the corresponding author upon reasonable request.

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
