# Peer review of "Fungicide Resistance in Botrytis spp. and Regional Strategies for Its Management in Northern European Strawberry Production"

_biotech, 2023, doi:10.3390/biotech12040064_

Round 1

Reviewer 1 Report

Comments and Suggestions for Authors

This review paper focuses on the management of fungicide resistance in the context of grey mould, a major disease affecting strawberries. It provides an in-depth analysis of the issue of fungicide resistance in the context of managing grey mold, in strawberry. It covers various aspects of the problem, including the development of resistance, the role of nursery plants, and non-chemical approaches, relying on empirical evidence from Northern German and Danish strawberry production, making it relevant and practical for the regions it discusses. The inclusion of specific data and figures strengthens the review's credibility.

Also, this review offers few practical recommendations for managing fungicide resistance, including strategies for strawberry farmers, the importance of nursery plant quality control, and the potential use of biological control agents.

Nonetheless, there are several weaknesses of this review that should be addressed and/or corrected. The paper is heavily focused on Northern German and Danish strawberry production, which limits its generalizability to other regions. While it mentions that similar trends are observed worldwide, more comparative analysis and data from diverse regions could enhance the paper's relevance to a broader audience. Otherwise, the authors must emphasize these regions in the title and abstract.

To increase the review's applicability to a broader audience, provide more context and data from regions outside Northern Germany and Denmark. Discuss how the findings might apply to different climates, strawberry varieties, and farming practices.

Importantly, this review could benefit from a more explicit differentiation from other similar studies on Botrytisresistance management (from the same authors or others). By clearly highlighting what sets this review apart in terms of methodology, scope, geographical focus (Germany and Denmark), or emphasis on practical case studies, the authors can better establish the uniqueness and significance of their work.

While the paper touches on fungicide resistance, it might be considered a weakness if it lacks an in-depth exploration of the mechanisms of resistance in field strains of Botrytis cinerea. A more detailed discussion of the molecular and genetic mechanisms behind resistance could enhance the understanding of how resistance develops and is sustained in nature.

Besides, the paper contains a substantial amount of information, and the organization could be improved to make it more reader-friendly. Narrowing-down the already (well) known information and breaking down complex sections into subsections (+ providing clear headings) could improve the paper's flow.

Importantly, while the paper briefly discusses non-chemical approaches/alternatives, it could benefit from a more detailed exploration of the efficacy of BCA and other non-chemical methods and any ongoing research in this area. This would provide a more balanced view of potential and sustainable solutions.

Ensure that the paper is written in a clear and concise manner, avoiding overly technical jargon and providing explanations where necessary.

Additional comments:

-       The title: The term "integrated approach" is not appropriate since the review paper does not cover a range of strategies and considerations for managing fungicide resistance. The title should effectively communicate the paper's subject matter to readers. (i.e. Fungicide Resistance Management Strategies for Controlling Botrytis spp. in Northern European Strawberry Production: Case Studies from Germany and Denmark).

-       The integration of a section focusing on precision agriculture and advanced data analytics. This could involve the use of sensors, remote sensing technologies, and artificial intelligence to optimize fungicide application could be briefly discussed in the text. 

-       Authors must consider addressing the potential ecological impacts of different control strategies, including the use of chemicals agents. This could include unintended consequences for non-target species, soil health, and overall ecosystem balance. Discussing these ecological aspects could highlight potential weaknesses in various Botrytis management approaches.

By addressing the points above, this review paper may become more accessible, relevant, and informative for a wider audience of researchers and practitioners in the field of horticulture and fungicide resistance management.

Comments on the Quality of English Language

The English -in most of the text- reads well, but the authors could shorten some of the sentences.

Reviewer 2 Report

Comments and Suggestions for Authors

The manuscript submitted for review is a typical review work. The authors undertook to develop an important topic from the point of view of the agricultural industry, i.e. "An integrated approach to managing fungicide resistance in Botrytis spp. in Northern European strawberry production". In my opinion, the authors fulfilled their task well. The work is written in an interesting way and well designed graphically. I am aware that there is always something to add and correct in the manuscript, but I think that this work is complete and can be accepted for publication in its current form. Good job.

Comments on the Quality of English Language

The manuscript looks good in terms of linguistic correctness. Minor adjustments may need to be made.

Author Response

We take comfort from this feedback because this is exactly what we want to achieve – a complete, readable, pragmatic, regional account of a complex problem.

We have adjusted the writing, as also suggested by Revviewer 1.

Reviewer 3 Report

Comments and Suggestions for Authors

The manuscript titled "An integrated approach to managing fungicide resistance in Botrytis spp. in Northern European strawberry production" is a review article on gray rot caused by Botrytis spp., which is the leading cause of fruit rot in strawberries and other fruit crops worldwide. Based on research conducted in Northern and Danish strawberry production, the authors developed a concept for managing fungicide resistance. They showed that practical implementation of such a strategy in northern Germany and Denmark reduces the occurrence of multi-resistant strains to a tolerable steady-state level.

The manuscript submitted for evaluation is interesting and has the potential to interest readers. It is an important contribution to the field of horticulture, especially strawberry production. The manuscript is well constructed and well-illustrated, which increases its value and interest among readers. It is prepared based on well-selected literature on the presented topic. You can further increase the value of a review article by adding a "Conclusions" chapter at the end of the manuscript, in which the authors present the most important postulates, point out weaknesses and, what is very important, present guidelines for further research. Furthermore, please, be sure that all the references cited in the manuscript are also included in the reference list and vice versa with matching spellings and dates.

Author Response

We have added a Conclusions section as suggested. This rounds off the article rather nicely. We have repeatedly cross-checked all references cited and have eliminated all mistakes and mismatches which we found. 

Round 2

Reviewer 1 Report

Comments and Suggestions for Authors

The authors addressed most of the comments.